# An Architecture for Deep, Hierarchical Generative Models

**Philip Bachman**
phil.bachman@maluuba.com
Maluuba Research

## Abstract

We present an architecture which lets us train deep, directed generative models with many layers of latent variables. We include deterministic paths between all latent variables and the generated output, and provide a richer set of connections between computations for inference and generation, which enables more effective communication of information throughout the model during training. To improve performance on natural images, we incorporate a lightweight autoregressive model in the reconstruction distribution. These techniques permit end-to-end training of models with 10+ layers of latent variables. Experiments show that our approach achieves state-of-the-art performance on standard image modelling benchmarks, can expose latent class structure in the absence of label information, and can provide convincing imputations of occluded regions in natural images.

## 1 Introduction

Training deep, directed generative models with many layers of latent variables poses a challenging problem. Each layer of latent variables introduces variance into gradient estimation which, given current training methods, tends to impede the flow of subtle information about sophisticated structure in the target distribution. Yet, for a generative model to learn effectively, this information needs to propagate from the terminal end of a stochastic computation graph, back to latent variables whose effect on the generated data may be obscured by many intervening sampling steps.

One approach to solving this problem is to use recurrent, sequential stochastic generative processes with strong interactions between their inference and generation mechanisms, as introduced in the DRAW model of Gregor et al. [5] and explored further in [1, 19, 22]. Another effective technique is to use lateral connections for merging bottom-up and top-down information in encoder/decoder type models. This approach is exemplified by the Ladder Network of Rasmus et al. [17], and has been developed further for, e.g. generative modelling and image processing in [8, 23].

Models like DRAW owe much of their success to two key properties: they decompose the process of generating data into many small steps of iterative refinement, and their structure includes direct deterministic paths between all latent variables and the final output. In parallel, models with lateral connections permit different components of a model to operate at well-separated levels of abstraction, thus generating a hierarchy of representations. This property is not explicitly shared by DRAW-like models, which typically reuse the same set of latent variables throughout the generative process. This makes it difficult for any of the latent variables, or steps in the generative process, to individually capture abstract properties of the data. We distinguish between the depth used by DRAW and the depth made possible by lateral connections by describing them respectively as sequential depth and hierarchical depth. These two types of depth are complimentary, rather than competing.

Our contributions focus on increasing hierarchical depth without forfeiting trainability. We combine the benefits of DRAW-like models and Ladder Networks by developing a class of models which we

call *Matryoshka Networks* (abbr. MatNets), due to their deeply nested structure. In Section 2, we present the general architecture of a MatNet. In the MatNet architecture we:

- Combine the ability of, e.g. LapGANs [3] and Diffusion Nets [21] to learn hierarchically-deep generative models with the power of jointly-trained inference/generation[1].

- Use lateral connections, shortcut connections, and residual connections [7] to provide direct paths through the inference network to the latent variables, and from the latent variables to the generated output — this makes hierarchically-deep models easily trainable in practice.

Section 2 also presents several extensions to the core architecture including: mixture-based prior distributions, a method for regularizing inference to prevent overfitting in practical settings, and a method for modelling the reconstruction distribution $p(x|z)$ with a lightweight, local autoregressive model. In Section 3, we present experiments showing that MatNets offer state-of-the-art performance on standard benchmarks for modelling simple images and compelling qualitative performance on challenging imputation problems for natural images. Finally, in Section 4 we provide further discussion of related work and promising directions for future work.

## 2 The Matryoshka Network Architecture

Matryoshka Networks combine three components: a *top-down network* (abbr. TD network), a *bottom-up network* (abbr. BU network), and a set of *merge modules* which merge information from the BU and TD networks. In the context of stochastic variational inference [10], all three components contribute to the approximate posterior distributions used during inference/training, but only the TD network participates in generation. We first describe the MatNet model formally, and then provide a procedural description of its three components. The full architecture is summarized in Fig. 1.

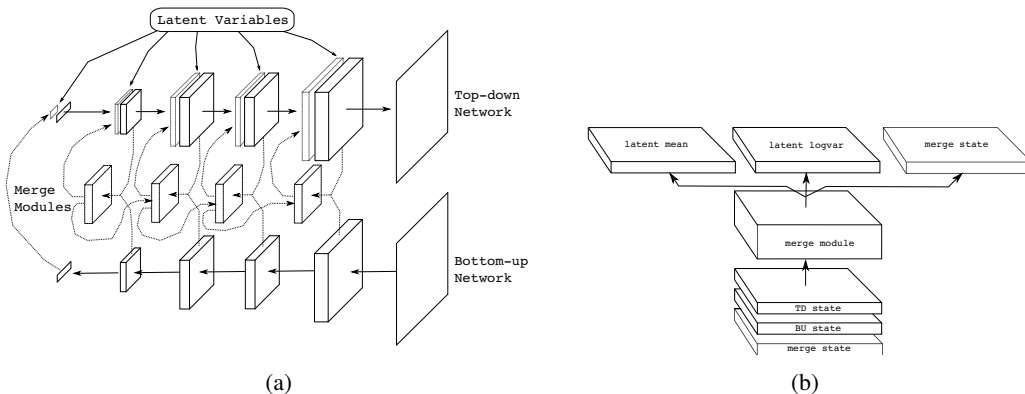

(a)                                        (b)

Figure 1: (a) The overall structure of a Matryoshka Network, and how information flows through the network during training. First, we perform a feedforward pass through the bottom-up network to generate a sequence of BU states. Next, we sample the initial latent variables conditioned on the final BU state. We then begin a stochastic feedforward pass through the top-down network. Whenever this feedforward pass requires sampling some latent variables, we get the sampling distribution by passing the corresponding TD and BU states through a merge module. This module draws conditional samples of the latent variables via reparametrization [10]. These latent samples are then combined with the current TD state, and the feedforward pass continues. Intuitively, this approach allows the TD network to invert the bottom-up network by tracking back along its intermediate states, and eventually recover its original input. (b) Detailed view of a merge module from the network in (a). This module stacks the relevant BU, TD, and merge states on top of each other, and then passes them through a convolutional residual module, as described in Eqn. 10. The output has three parts — the first provides means for the latent variables, the second provides their log-variances, and the third conveys updated state information to subsequent merge modules.

## 2.1 Formal Description

The distribution $p(x)$ generated by a MatNet is encoded in its top-down network. To model $p(x)$, the TD network decomposes the joint distribution $p(x, \mathbf{z})$ over an observation $x$ and a sequence of latent variables $\mathbf{z} \equiv \{z_0, ..., z_d\}$ into a sequence of simpler conditional distributions:

$$p(x) = \sum_{(z_d, ..., z_0)} p(x|z_d, ..., z_0)p(z_d|z_{d-1}, ..., z_0)...p(z_i|z_{i-1}, ..., z_0)...p(z_0), \qquad (1)$$

which we marginalize with respect to the latent variables to get $p(x)$. The TD network is designed so that each conditional $p(z_i|z_{i-1}, ..., z_0)$ can be truncated to $p(x|h_i^t)$ using an internal *TD state* $h_i^t$. See Eqns. 7/8 in Sec. 2.2 for procedural details.

The distribution $q(\mathbf{z}|x)$ used for inference in an unconditional MatNet involves the BU network, TD network, and merge modules. This distribution can be written:

$$q(z_d, ..., z_0|x) = q(z_0|x)q(z_1|z_0, x)...q(z_i|z_{i-1}, ..., z_0, x)...q(z_d|z_{d-1}, ..., z_0, x), \qquad (2)$$

where each conditional $q(z_i|z_{i-1}, ..., z_0, x)$ can be truncated to $q(z_i|h_{i+1}^m)$ using an internal *merge state* $h_{i+1}^m$ produced by the $i$th merge module. See Eqns. 10/11 in Sec. 2.2 for procedural details.

MatNets can also be applied to conditional generation problems like inpainting or pixel-wise segmentation. For, e.g. inpainting with known pixels $x^k$ and missing pixels $x^u$, the predictive distribution of a conditional MatNet is given by:

$$p(x^u|x^k) = \sum_{(z_d, ..., z_0)} p(x^u|z_d, ..., z_0, x^k)p(z_d|z_{d-1}, ..., z_0, x^k)...p(z_1|z_0, x^k)p(z_0|x^k). \qquad (3)$$

Each conditional $p(z_i|z_{i-1}, ..., z_0, x^k)$ can be truncated to $p(z_i|h_{i+1}^{m:g})$, where $h_{i+1}^{m:g}$ indicates state in a merge module belonging to the generator network. Crucially, conditional MatNets include BU networks and merge modules that participate in generation, in addition to the BU networks and merge modules used by both conditional and unconditional MatNets during inference/training.

The distribution used for inference in a conditional MatNet is given by:

$$q(z_d, ..., z_0|x^k, x^u) = q(z_d|z_{d-1}, ..., z_0, x^k, x^u)...q(z_1|z_0, x^k, x^u)q(z_0|x^k, x^u), \qquad (4)$$

where each conditional $q(z_i|z_{i-1}, ..., z_0, x^k, x^u)$ can be truncated to $q(z_i|h_{i+1}^{m:i})$, where $h_{i+1}^{m:i}$ indicates state in a merge module belonging to the inference network. Note that, in a conditional MatNet the distributions $p(\cdot|\cdot)$ are not allowed to condition on $x^u$, while the distributions $q(\cdot|\cdot)$ can.

MatNets are well-suited to training with Stochastic Gradient Variational Bayes [10]. In SGVB, one maximizes a lower-bound on the data log-likelihood based on the variational free-energy:

$$\log p(x) \geq \mathbb{E}_{\mathbf{z} \sim q(\mathbf{z}|x)} \left[\log p(x|\mathbf{z})\right] - \mathrm{KL}(q(\mathbf{z}|x) \,||\, p(\mathbf{z})), \qquad (5)$$

for which $p$ and $q$ must satisfy a few simple assumptions and $\mathrm{KL}(q(\mathbf{z}|x) \,||\, p(\mathbf{z}))$ indicates the KL divergence between the inference distribution $q(\mathbf{z}|x)$ and the model prior $p(\mathbf{z})$. This bound is tight when the inference distribution matches the true posterior $p(\mathbf{z}|x)$ in the model joint distribution $p(x, \mathbf{z}) = p(x|\mathbf{z})p(\mathbf{z})$ — in our case given by Eqns. 1/3.

For brevity, we only explicitly write the free-energy bound for a conditional MatNet, which is:

$$\log p(x^u|x^k) \geq \mathbb{E}_{q(z_d, ..., z_0|x^k, x^u)} \left[\log p(x^u|z_d, ..., z_0, x^k)\right] - \qquad (6)$$
$$\mathrm{KL}(q(z_d, ..., z_0|x^k, x^u)||p(z_d, ..., z_0|x^k)).$$

With SGVB we can optimize the bound in Eqn. 6 using the "reparametrization trick" to allow easy backpropagation through the expectation over $\mathbf{z} \sim q(\mathbf{z}|x^k, x^u)$. See [10, 18] for more details about this technique. The bound for unconditional MatNets is nearly identical — it just removes $x^k$.

## 2.2 Procedural Description

Structurally, top-down networks in MatNets comprise sequences of modules in which each module $f_i^t$ receives two inputs: a deterministic top-down state $h_i^t$ from the preceding module $f_{i-1}^t$, and some

latent variables $z_i$. Module $f_i^t$ produces an updated state $h_{i+1}^t = f_i^t(h_i^t, z_i; \theta^t)$, where $\theta^t$ indicates the TD network's parameters. By defining the TD modules appropriately, we can reproduce the architectures for LapGANs, Diffusion Nets, and Probabilistic Ladder Networks [23]. Motivated by the success of LapGANs and ResNets [7], we use TD modules in which the latent variables are concatenated with the top-down state, then transformed, after which the transformed values are added back to the top-down state prior to further processing. If the adding occurs immediately before, e.g. a ReLU, then the latent variables can effectively gate the top-down state by knocking particular elements below zero. This allows each stochastic module in the top-down network to apply small refinements to the output of preceding modules. MatNets thus perform iterative stochastic refinement through hierarchical depth, rather than through sequential depth as in DRAW[2].

More precisely, the top-down modules in our convolutional MatNets compute:

$$h_{i+1}^t = \mathrm{lrelu}(h_i^t + \mathrm{conv}(\mathrm{lrelu}(\mathrm{conv}([h_i^t; z_i], v_i^t)), w_i^t)), \qquad (7)$$

where $[x; x']$ indicates tensor concatenation along the "feature" axis, $\mathrm{lrelu}(\cdot)$ indicates the leaky ReLU function, $\mathrm{conv}(h, w)$ indicates shape-preserving convolution of the input $h$ with the kernel $w$, and $w_i^t/v_i^t$ indicate the trainable parameters for module $i$ in the TD network. We elide bias terms for brevity. When working with fully-connected models we use stochastic GRU-style state updates rather than the stochastic residual updates in Eq. 7. Exhaustive descriptions of the modules can be found in our code at: `https://github.com/Philip-Bachman/MatNets-NIPS`.

These TD modules represent each conditional $p(z_i|z_{i-1}, ..., z_0)$ in Eq. 1 using $p(z_i|h_i^t)$. TD module $f_i^t$ places a distribution over $z_i$ using parameters $[\bar{\mu}_i; \log \bar{\sigma}_i^2]$ computed as follows:

$$[\bar{\mu}_i; \log \bar{\sigma}_i^2] = \mathrm{conv}(\mathrm{lrelu}(\mathrm{conv}(h_i^t, v_i^t)), w_i^t), \qquad (8)$$

where we use $\bar{\cdot}$ to distinguish between Gaussian parameters from the generator network and those from the inference network (see Eqn. 11). The distributions $p(\cdot)$ all depend on the parameters $\theta^t$.

Bottom-up networks in MatNets comprise sequences of modules in which each module receives input only from the preceding BU module. Our BU networks are all deterministic and feedforward, but sensibly augmenting them with auxiliary latent variables [16, 15] and/or recurrence is a promising topic for future work. Each non-terminal module $f_i^b$ in the BU network computes an updated state: $h_i^b = f_i^b(h_{i+1}^b; \theta^b)$. The final module, $f_0^b$, provides means and log-variances for sampling $z_0$ via reparametrization [10]. To align BU modules with their counterparts in the TD network, we number them in reverse order of evaluation. We structured the modules in our BU networks to take advantage of residual connections. Specifically, each BU module $f_i^b$ computes:

$$h_i^b = \mathrm{lrelu}(h_{i+1}^b + \mathrm{conv}(\mathrm{lrelu}(\mathrm{conv}(h_{i+1}^b, v_i^b)), w_i^b)), \qquad (9)$$

with operations defined as for Eq. 7. These updates can be replaced by GRUs, LSTMs, etc.

The updates described in Eqns. 7 and 9 both assume that module inputs and outputs are the same shape. We thus construct MatNets using groups of "meta modules", within which module input/output shapes are constant. To keep our network design (relatively) simple, we use one meta module for each spatial scale in our networks (e.g. scales of 14x14, 7x7, and fully-connected for MNIST). We connect meta modules using layers which may upsample, downsample, and change feature dimension via strided convolution. We use standard convolution layers, possibly with up or downsampling, to feed data into and out of the bottom-up and top-down networks.

During inference, merge modules compare the current top-down state with the state of the corresponding bottom-up module, conditioned on the current merge state, and choose a perturbation of the top-down information to push it towards recovering the bottom-up network's input (i.e. minimize reconstruction error). The $i$th merge module outputs $[\mu_i; \log \sigma_i^2; h_{i+1}^m] = f_i^m(h_i^b, h_i^t, h_i^m; \theta^m)$, where $\mu_i$ and $\log \sigma_i^2$ are the mean and log-variance for sampling $z_i$ via reparametrization, and $h_{i+1}^m$ gives the updated merge state. As in the TD and BU networks, we use a residual update:

$$h_{i+1}^m = \mathrm{lrelu}(h_i^m + \mathrm{conv}(\mathrm{lrelu}(\mathrm{conv}([h_i^m; h_i^b; h_i^t], u_i^i)), v_i^i)) \qquad (10)$$

$$[\mu_i; \log \sigma_i^2] = \mathrm{conv}(h_{i+1}^m, w_i^i), \qquad (11)$$

in which the convolution kernels $u_i^i$, $v_i^i$, and $w_i^i$ constitute the trainable parameters of this module. Each merge module thus computes an updated merge state and then reparametrizes a diagonal Gaussian using a linear function of the updated merge state.

In our experiments all modules in all networks had their own trainable parameters. We experimented with parameter sharing and GRU-style state in our convolutional models. The stochastic convolutional GRU is particularly interesting when applied depth-wise (rather than time-wise as in [19]), as it implements a stochastic Neural GPU [9] trainable by variational inference and capable of multi-modal dynamics. We saw no performance gains with these changes, but they merit further investigation.

In unconditional MatNets, the top-most latent variables $z_0$ follow a zero-mean, unit-variance Gaussian prior, except in our experiments with mixture-based priors. In conditional MatNets, $z_0$ follows a distribution conditioned on the known values $x^k$. Conditional MatNets use parallel sets of BU and merge modules for the conditional generator and the inference network. BU modules in the conditional generator observe a partial input $x^k$, while BU modules in the inference network observe both $x^k$ and the unknown values $x^u$ (which the model is trained to predict). The generative BU and merge modules in a conditional MatNet interact with the TD modules analogously to the BU and merge modules used for inference. Our models used independent Bernoullis, diagonal Gaussians, or "integrated" Logistics (see [11]) for the final output distribution $p(x|z_d, ..., z_0)/p(x^u|z_d, ..., z_0, x^k)$.

## 2.3 Model Extensions

We also develop several extensions for the MatNet architecture. The first is to replace the zero-mean, unit-variance Gaussian prior over $z_0$ with a Gaussian Mixture Model, which we train simultaneously with the rest of the model. When using a mixture prior, we use an analytical approximation to the required KL divergence. For Gaussian distribution $q$, and Gaussian mixture $p$ with components $\{p_1, ..., p_k\}$ with uniform mixture weights, we use the KL approximation:

$$\mathrm{KL}(q \,\|\, p) \approx \log \frac{1}{\sum_{i=1}^{k} e^{-\,\mathrm{KL}(q \,\|\, p_i)}}. \tag{12}$$

Our tests with mixture-based priors are only concerned with qualitative behaviour, so we do not worry about the approximation error in Eqn. 12.

The second extension is a technique for regularizing the inference model to prevent overfitting beyond that which is present in the generator. This regularization is applied by optimizing:

$$\underset{q}{\text{maximize}} \; \underset{x \sim p(x)}{\mathbb{E}} \left[ \underset{\mathbf{z} \sim q(\mathbf{z}|x)}{\mathbb{E}} \left[ \log p(x|\mathbf{z}) \right] - \mathrm{KL}(q(\mathbf{z}|x) \,\|\, p(\mathbf{z})) \right]. \tag{13}$$

This maximizes the free-energy bound for samples drawn from our model, but without changing their true log-likelihood. By maximizing Eqn. 13, we implicitly reduce $\mathrm{KL}(q(\mathbf{z}|x) \,\|\, p(\mathbf{z}|x))$, which is the gap between the free-energy bound and the true log-likelihood. A similar regularizer can be constructed for minimizing $\mathrm{KL}(p(\mathbf{z}|x) \,\|\, q(\mathbf{z}|x))$. We use (13) to reduce overfitting, and slightly boost test performance, in our experiments with MNIST and Omniglot.

The third extension off-loads responsibility for modelling sharp local dynamics in images, e.g. precise edge placements and small variations in textures, from the latent variables onto a local, deterministic autoregressive model. We use a simplified version of the masked convolutions in the PixelCNN of [25], modified to condition on the output of the final TD module in a MatNet. This modification is easy — we just concatenate the final TD module's output and the true image, and feed this into a PixelCNN with, e.g. five layers. A trick we use to improve gradient flow back to the MatNet is to feed the MatNet's output directly into each internal layer of the PixelCNN. In the masked convolution layers, connections to the MatNet output are unrestricted, since they are already separated from the ground truth by an appropriately-monitored noisy channel. Larger, more powerful mechanisms for combining local autoregressions and conditioning information are explored in [26].

## 3 Experiments

We measured quantitative performance of MatNets on three datasets: MNIST, Omniglot [13], and CIFAR 10 [12]. We used the 28x28 version of Omniglot described in [2], which can be found at: `https://github.com/yburda/iwae`. All quantitative experiments measured performance in

| Model | Test NLL |
|---|---|
| VAE (2 layers, 5 samples) | 106.31 |
| IWAE (2 layers, 50 samples) | 103.38 |
| RBM (500 hidden) | 100.46 |
| DRAW | < 96.5 |
| Conv DRAW | < 91.0 |
| MatNet | < 89.5 |

Table 1: NLL on 28x28 Omniglot.

| Model | Test NLL |
|---|---|
| Uniform Distribution | 8.00 |
| Multivariate Gaussian | 4.70 |
| NICE [4] | 4.48 |
| Deep Diffusion [22] | 4.20 |
| Deep GMMs [26] | 4.00 |
| Pixel RNN [27] | 3.00 |
| DRAW | < 4.13 |
| Conv DRAW | < 3.58 |
| MatNet | < 3.68 |
| MatNet+AR | < 3.24 |

Table 2: NLL on CIFAR 10.

| Model | Test NLL |
|---|---|
| DRAW (64 steps, Gaussian attention) | ≤ 80.9 |
| DRAW (80 steps, ST att and CGRU canvas) | ≤ 80.5 |
| Pixel RNN | ≤ 79.2 |
| MatNet | ≤ 78.5 |

Table 3: NLL on MNIST – Convolutional.

| Model | Test NLL |
|---|---|
| DRAW (16 steps, no attention) [8] | ≤ 87.4 |
| VAE + IWAE [2] | ≤ 82.9 |
| Prob Ladder [28] | ≤ 81.2 |
| FC MatNet | ≤ 80.5 |

Table 4: NLL on MNIST – Fully-connected.

| Method | 1 quadrant | 2 quadrants | 3 quadrants |
|---|---|---|---|
| NN | 99.75 | 62.18 | 25.99 |
| GSNN | 99.82 | 62.41 | 26.29 |
| CVAE | 68.39 | 45.34 | 20.96 |
| CVAE | 63.91 | 44.73 | 20.95 |
| MatNet | 57.45 | 36.82 | 17.96 |

Structured prediction NLL with $n$ quadrants visible [27]

Figure 2: MatNet performance on quantitative benchmarks. All tables except the lower-right table describe standard unconditional generative NLL results. The lower-right table presents results from the structured prediction task in [22], in which 1-3 quadrants of an MNIST digit are visible, and NLL is measured on predictions for the unobserved quadrants.

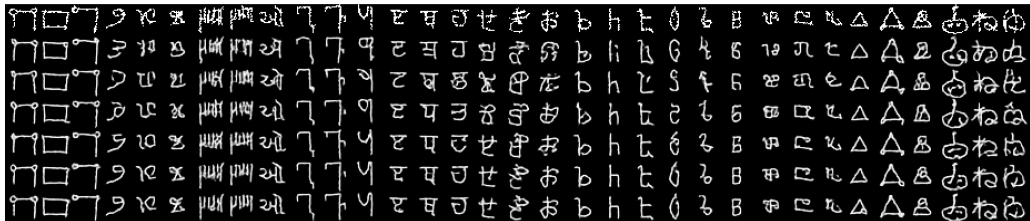

Figure 3: Class-like structure learned by a MatNet trained on 28x28 Omniglot, without label information. The model used a GMM prior over $z_0$ with 50 mixture components. Each group of three columns corresponds to a mixture component. The top row shows validation set examples whose posterior over the mixture components placed them into each component. Subsequent rows show samples drawn by freely resampling latent variables from the model prior, conditioned on the top $k$ layers of latent variables, i.e. $\{z_0, ..., z_{k-1}\}$ being drawn from the approximate posterior for the example at the top of the column. From the second row down, we show $k = \{1, 2, 4, 6, 8, 10\}$.

terms of negative log-likelihood, with the CIFAR 10 scores rescaled to bits-per-pixel and corrected for discrete/continuous observations as described in [24]. We used the IWAE bound from [2] to evaluate our models, with 2500 samples in the bound. We performed additional experiments measuring the qualitative performance of MatNets using Omniglot, CelebA faces [14], LSUN 2015 towers, and LSUN 2015 churches. The latter three datasets are 64x64 color images with significant detail and non-trivial structure. Complete hyperparameters for model architecture and optimization can be found in the code at `https://github.com/Philip-Bachman/MatNets-NIPS`.

We performed three quantitative tests using MNIST. The first tests measured generative performance on dynamically-binarized images using a fully-connected model (for comparison with [2, 23]) and on the fixed binarization from [20] using a convolutional model (for comparison with [25, 19]). MatNets improved on existing results in both settings. See the tables in Fig. 2. Our third tests with MNIST measured performance of conditional MatNets for structured prediction. For this, we recreated the tests described in [22]. MatNet performance on these tests was also strong, though the prior results were from a fully-connected model, which skews the comparison.

We also measured quantitative performance using the 32x32 color images of CIFAR 10. We trained two models on this data — one with a Gaussian reconstruction distribution and dequantization as described in [24], and the other which added a local autoregression and used the "integrated Logistic" likelihood described in [11]. The Gaussian model fell just short of the best previously reported result for a variational method (from [6]), and well short of the Pixel RNN presented in [25]. Performance on this task seems very dependent on a model's ability to predict pixel intensities precisely along edges. The ability to efficiently capture global structure has a relatively weak benefit. Mistaking a cat for a dog costs little when amortized over thousands of pixels, while misplacing a single edge can spike the reconstruction cost dramatically. We demonstrate the strength of this effect in Fig. 4, where we plot how the bits paid to encode observations are distributed among the modules in the network over the course of training for MNIST, Omniglot, and CIFAR 10. The plots show a stark difference between these distributions when modelling simple line drawings vs. when modelling more natural

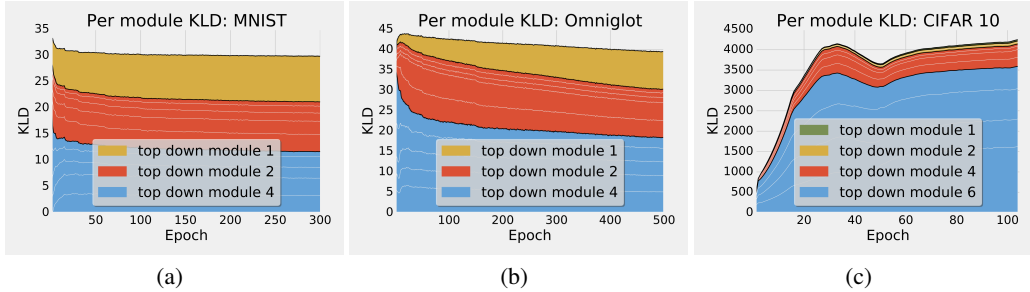

|     |     |     |
| :-: | :-: | :-: |
| (a) | (b) | (c) |

Figure 4: This figure shows per module divergences $\mathrm{KL}(q(z_i|h_{i+1}^m) \, || \, p(z_i|h_i^t))$ over the course of training for models trained on MNIST, Omniglot, and CIFAR 10. The stacked area plots are grouped by "meta module" in the TD network. The MNIST and Omniglot models both had a single FC module and meta modules at spatial dimension 7x7 and 14x14. The meta modules at 7x7 and 14x14 both comprised 5 TD modules. The CIFAR10 model (without autoregression) had one FC module, and meta modules at spatial dimension 8x8, 16x16, and 32x32. These meta modules comprised 2, 4, and 4 modules respectively. Light lines separate modules, and dark lines separate meta modules. The encoding cost on CIFAR 10 is clearly dominated by the low-level details encoded by the latent variables in the full-resolution TD modules closest to the output.

images. For CIFAR 10, almost all of the encoding cost was spent in the 32x32 layers of the network closest to the generated output. This was our motivation for adding a lightweight autoregression to $p(x|z)$, which significantly reduced the gap between our model and the PixelRNN. Fig. 5 shows some samples from our model, which exhibit occasional glimpses of global and local structure.

Our final quantitative test used the Omniglot handwritten character dataset, rescaled to 28x28 as in [2]. These tests used the same convolutional architecture as on MNIST. Our model outperformed previous results, as shown in Fig. 2. Using Omniglot we also experimented with placing a mixture-based prior distribution over the top-most latent variables $z_0$. The purpose of these tests was to determine whether the model could uncover latent class structure in the data without seeing any label information. We visualize results of these tests in Fig. 3. Additional description is provided in the figure caption. We placed a slight penalty on the entropy of the posterior distributions for each input to the model, to encourage a stronger separation of the mixture components. The inputs assigned to each mixture component (based on their posteriors) exhibit clear stylistic coherence.

In addition to qualitative tests exploring our model's ability to uncover latent factors of variation in Omniglot data, we tested the performance of our models at imputing missing regions of higher resolution images. These tests used images of celebrity faces, churches, and towers. These images include far more detail and variation than those in MNIST/Omniglot/CIFAR 10. We used two-stage models for these tests, in which each stage was a conditional MatNet. The first stage formed an initial guess for the missing image content, and the second stage then refined that guess. Both stages used the same architectures for their inference and generator networks. We sampled imputation problems by placing three 20x20 occluders uniformly at random in the image. Each stage had single TD modules at scales 32x32, 16x16, 8x8, and fully-connected. We trained models for roughly 200k updates, and show imputation performance on images from a test set that was held out during training. Results are shown in Fig. 5.

## 4 Related Work and Discussion

Previous successful attempts to train hierarchically-deep models largely fall into a class of methods based on deconstructing, and then reconstructing data. Such approaches are akin to solving mazes by starting at the end and working backwards, or to learning how an object works by repeatedly disassembling and reassembling it. Examples include LapGANs [3], which deconstruct an image by repeatedly downsampling it, and Diffusion Nets [21], which deconstruct arbitrary data by subjecting it to a long sequence of small random perturbations. The power of these approaches stems from the way in which gradually deconstructing the data leaves behind a trail of crumbs which can be followed back to a well-formed observation. In the generative models of [3, 21], the deconstruction processes were defined a priori, which avoided the need for trained inference. This makes training significantly

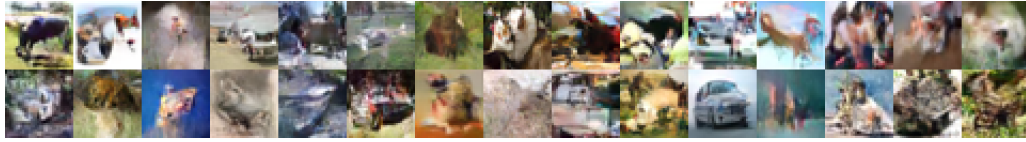

(a) CIFAR 10 samples

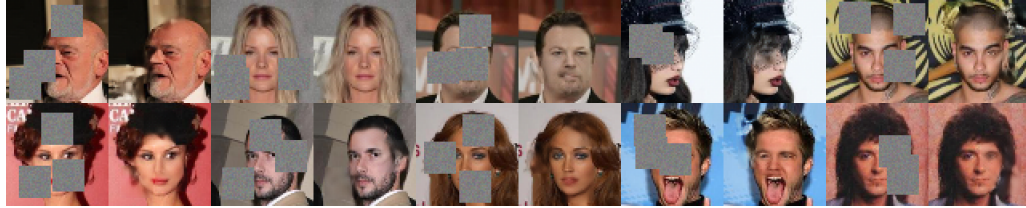

(b) CelebA Faces

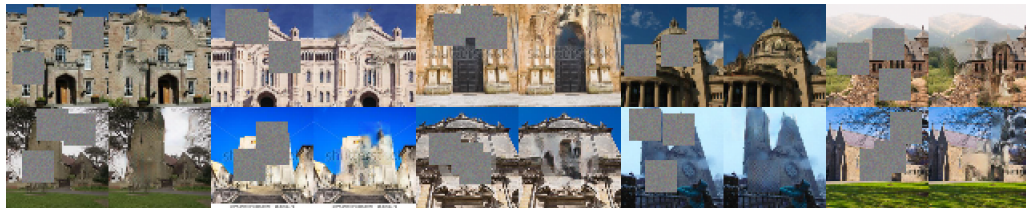

(c) LSUN Churches

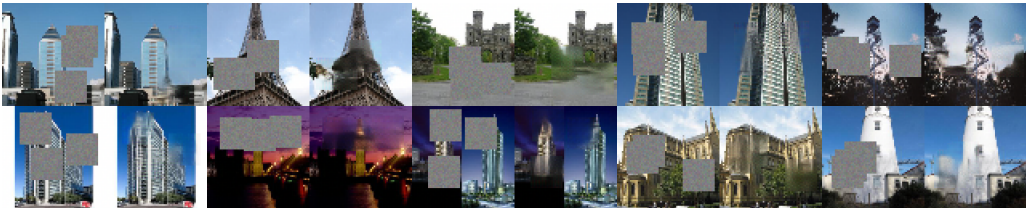

(d) LSUN Towers

Figure 5: Imputation results on challenging, real-world images. These images show predictions for missing data generated by a two stage conditional MatNet, trained as described in Section 3. Each occluded region was 20x20 pixels. Locations for the occlusions were selected uniformly at random within the images. One interesting behaviour which emerged in these tests was that our model successfully learned to properly reconstruct the watermark for "shutterstock", which was a source of many of the LSUN images – see the second input/output pair in the third row of (b).

easier, but subverts one of the main motivations for working with latent variables and sample-based approximate inference, i.e. the ability to capture salient factors of variation in the inferred relations between latent variables and observed data. This deficiency is beginning to be addressed by, e.g. the Probabilistic Ladder Networks of [23], which are a special case of our architecture in which the deterministic paths from latent variables to observations are removed and the conditioning mechanism in inference is more restricted.

Reasoning about data through the posteriors induced by an appropriate generative model motivates some intriguing work at the intersection of machine learning and cognitive science. This work shows that, in the context of an appropriate generative model, powerful inference mechanisms are capable of exposing the underlying factors of variation in fairly sophisticated data. See, e.g. Lake et al. [13]. Techniques for training coupled generation and inference have now reached a level that makes it possible to investigate these ideas while learning models end-to-end [4].

In future work we plan to apply our models to more "interesting" generative modelling problems, including more challenging image data and problems in language/sequence modelling. The strong performance of our models on benchmark problems suggests their potential for solving difficult structured prediction problems. Combining the hierarchical depth of MatNets with the sequential depth of DRAW is also worthwhile.

## Footnotes

[1]A significant downside of LapGANs and Diffusion Nets is that they define their inference mechanisms a priori. This is computationally convenient, but prevents the model from learning abstract representations.

[2]Current DRAW-like models can be extended to incorporate hierarchical depth, and our models can be extended to incorporate sequential depth.

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
