[Reviews · NeurIPS 2016]

Reviewer 1

Summary

Arguing that the success of DRAW-like models is due their incremental approach to generating data combined with deterministic connections present between each latent variable and the observation, the authors apply the same design principles to hierarchical latent variable models. The resulting models, called Matryoshka Networks (MatNets), consist of a bottom-up (BU) network, top-down network (TD), and merge modules that directs information flow between the two. The MatNet architecture is reminiscent of Probabilistic Ladder Networks (PLNs) [28], with a deterministic pathway from the observation to a latent layers in the inference model and a top-down factorization of the variational posterior. The primary difference are deterministic connections between the latents and the observation in the Matryoshka Network, which are not present in PLNs. MatNets are evaluated at density modeling on MNIST, Omniglot, and CIFAR achieving either highly competitive or state-of-the-art results.

Qualitative Assessment

This is an interesting and fairly well executed paper. The main contribution of the paper is a hierarchical VAE architecture with deterministic connections in the inference and generative models, implementing incremental generation of observations. A similar approach has been pioneered by DRAW, but in DRAW-like models the layers of latent variables don't form a hierarchy (and are all at essentially the same level). Prob Ladder Nets have all the proposed features except for the deterministic connections in the generative model. Thus the main novel contribution here is the introduction of deterministic connections in a generative hierarchical model. Surprisingly, the experimental section has no experiments demonstrating the importance of deterministic connections in the inference model and/or the generative model. Instead, the experiments show that the proposed architecture performs very well on several dataset, which though interesting but does not provide direct evidence for the importance of the paper's central claim. The paper is fairly well written, but the presentation of the proposed architecture is not clear enough to be easily accessible. It would be much easier to read if the high-level (probabilistic) description of the model came before the specification of the exact computations involved (i.e. Procedural Description in Sec. 2.1). In a few places the lack of citations can make some statements appear to be claims of novelty. For example, the discussion of using convolutional GRUs modules fails to mention the prior use of these in generative models in [24]. Given the many similarities with DRAW and PLNs, it is surprising that the related work section mentions neither. While the relation of PLNs to DRAW is well explained in the introduction, PLNs are not mentioned in the paper at all, which is something the authors should correct. Was the inference regularization technique from Section 2.3 used for any of the experiments? Typos: Equations 11 and 12 both should have log in front for p(x|z).

Confidence in this Review

3-Expert (read the paper in detail, know the area, quite certain of my opinion)


Reviewer 2

Summary

The authors present a new hierarchical generative model, the MatNet. It combines recent advances in the field by the DRAW-type models, Ladder Networks, residual nets, etc It achieves state-of-the-art results, in terms of NLL, in some datasets (MNIST, Omniglot, CIFAR 10), and exhibits some good qualitative inpainting results on other datasets (CelebA, LSUN).

Qualitative Assessment

Technical Quality: I thought the experimental section was very clear; a straightforward and reproducable experimental protocol was followed, and there were comparisons with state-of-the-art methods, even with some methods that were very recent (eg. Pixel RNN). I would have liked to see side-to-side comparison with other methods for the qualitative results which I did not see in the supplementary material either. Novelty: The authors used recent successes in the field and pushed forward eg by using a GMM as a prior over z_0 instead of a single Gaussian prior. Clarity: Figure 1 could have been of a much better quality and clearer. Also it was not clear what was different between the supplementary material and the actual paper submission (Except for the additional images section in the supp. material)

Confidence in this Review

1-Less confident (might not have understood significant parts)


Reviewer 3

Summary

The authors propose a an architecture for deep generative models that enable deeper models to be learned without forfeiting trainability. The improvement in trainability is believed to arise from the inclusion of lateral, shortcut, and residual connections within the network. The proposed architecture yields state-of-the-art performance on several benchmark datasets.

Qualitative Assessment

The authors provide reasonable justification for the architecture proposed, taking key insights from successes in recent work on deep recurrent networks and semi supervised ladder networks. The results reported are indeed impressive. While the premise of the paper is centered on the trainability of the proposed networks, evidence for their trainability comes only in the form of performance metrics on various benchmarks. If the trainability of these networks is a primary contribution of this work, further analysis thereof, or experiments specifically highlighting trainability would be desirable.

Confidence in this Review

1-Less confident (might not have understood significant parts)


Reviewer 4

Summary

A deep architecture for generative image models is developed, called MatNets. Performance is compared with the state of the art in terms of likelihoods.

Qualitative Assessment

Ideas from multiple successful generative image models are combined, leading to a rather complicated model. The obtained performance is good but only slightly better than the current state of the art.

Confidence in this Review

2-Confident (read it all; understood it all reasonably well)


Reviewer 5

Summary

The authors describe a method for training deep, generative models that allow for more information flow and more effective training. They present the Matryoshka network which bears resemblance to LapGANs and ResNets and comprises of a top-down network, a bottom-up network, and a set of merge modules. Merge modules compare the top-down states with the bottom-up module states, choosing a perturbation to minimize reconstruction error.

Qualitative Assessment

- While the details for specific sections are generally sufficient (such as good use of equations to clarify computations), there is room for improvement in the clarity and flow across different sections and subsections. - Section 2.2 and end of section 2.1 needs to connect the sampling/inference/distributions with the network components in a clearer manner. For example, x (in section 2.2) is never defined to be anything other than a random variable (until line 132, where it is vaguely defined as unknown or known data). How does this variable x relate to the network? - How are the latent variables z initialized? What kind of impact does this initialization have? - Is there a typo on line 71? It says that conv(h, w) is a convolution of the input x with kernel w, but says nothing about h. And, x isn't an input into the conv function. - The qualitative results are quite impressive. - The quantitative results show that the proposed model works better than previous methods, though it is much more complex. - Before diving into the details of individual components of the network, it would be helpful to provide a brief explanation on where the inputs and outputs of the network (and how they are used) are clearly defined. It would be even more helpful if the inputs and outputs of each component of the network (top-down, bottom-up, merge modules) are defined prior to explaining the mechanisms. - I would like to see more analysis on the qualitative results in addition to explanations about quantitative results. - Can you provide more measurements besides NLL? How about reconstruction error? - It would be great to see some qualitative comparisons with previous methods.

Confidence in this Review

1-Less confident (might not have understood significant parts)


Reviewer 6

Summary

This paper proposes a new class of architectures, named Matryoshka Networks (MatNets), which combine the benefits of DRAW-like models and Ladder Networks to learn hierarchically deep generative models with jointly-trained inference/generation. The paper shows the quantitative performance of MatNets on MNIST, Omniglot, and CIFAR10.

Qualitative Assessment

The paper proposes an interesting framework to learn deep generative models. The authors show nice results on imputing missing regions of images. However, it would be more convincing if the paper includes classification results (supervised/semi-supervised/unsupervised) on benchmarks such as MNIST and CIFAR10. The paper is well-written. The claims, explanation, and derivation in the paper are clear and easy to follow.

Confidence in this Review

1-Less confident (might not have understood significant parts)